# Does Additional Physical Education Improve Exam Performance at the End of Compulsory Education? A Secondary Analysis from a Natural Experiment: The CHAMPS-Study DK

**DOI:** 10.3390/children8010057

**Published:** 2021-01-18

**Authors:** Jakob Tarp, Anne Kær Gejl, Charles H. Hillman, Niels Wedderkopp, Anna Bugge

**Affiliations:** 1Department of Sports Medicine, Norwegian School of Sports Sciences, 0863 Oslo, Norway; 2Department of Midwifery, Physiotherapy, Occupational Therapy and Psychomotor Therapy, Faculty of Health, University College Copenhagen, 1799 Copenhagen, Denmark; annekaerg@gmail.com (A.K.G.); ABUG@kp.dk (A.B.); 3Department of Psychology, Department of Physical Therapy, Movement, & Rehabilitation Sciences, Northeastern University, Boston, MA 02115, USA; c.hillman@northeastern.edu; 4Department of Regional Health, University of Southern Denmark, 5000 Odense, Denmark; NWedderkopp@health.sdu.dk; 5Department of Orthopedics, University Hospital of South West Jutland, 6700 Esbjerg, Denmark

**Keywords:** school-based, intervention, policy, exercise, fitness

## Abstract

It remains unclear whether the provision of additional physical activity in school improves academic outcomes. We conducted a secondary analysis of the Childhood Health, Activity, and Motor Performance School Study Denmark (CHAMPS-study DK), a natural experiment based on a trebling of curricular physical education, to investigate whether children receiving additional physical education performed better on their academic exams at the conclusion of compulsory education (i.e., 9th grade). Children from six intervention schools received 3–7 years of exposure to 270 weekly minutes of physical education (sports schools), while children from four control schools received the 90-min national standard (normal schools). Academic performance was based on the standard Danish 7-point scale (ranging from −03 to 12) and retrieved from national registries. The primary outcome was calculated as the average exam grade. Comparisons of participants at sports and normal schools were adjusted for individual socioeconomic factors and school-level academic environment. There were no differences in the pooled exam performance among 691 sports- and 510 normal-school participants (0.20 (95% confidence interval: −0.12 to 0.52)). Results for subject-specific exams indicated similar results. This analysis from a non-randomized natural experiment did not provide evidence that simply adding additional physical education is sufficient to affect academic performance relative to the national standard.

## 1. Introduction

Regular physical activity carries a plethora of both physical and mental health benefits across the human lifespan [1]. Several lines of research further suggest that physical activity may be beneficial for the academic performance of school-age children [2,3,4,5]. Academic competencies such as mathematics and reading are highly important for individuals’ educational, vocational, economical, and overall life success [6,7,8]. Thus, the identification of factors supportive of academic achievement are important for the individual, and more broadly, society.

Intervention models for increasing in-school physical activity can roughly be divided into three categories: Physical activity delivered as part of the curriculum; that is, physical activity, for example, integrating into language, mathematics or science lessons. This can be delivered through tailored educational activities, often in the form of games or activities requiring bodily movement (e.g., running in triangles or jumping calculations on floor-paintings [9,10,11]. A second category provides physical activity in school yet separate from core academic subjects. This could be delivered via increasing or augmenting physical education (PE), providing active breaks during classes, or through programs such as “The Daily Mile” [12,13]. A third model would embrace both options aiming to improve the overall physical activity environment throughout the school day [14,15,16,17]. Exploring the effectiveness of these different models within the school system is needed to translate mechanistic studies into the “real-world” and to assess feasibility and efficacy of the potential policies needed to implement the intervention at scale. For example, it may be easier to implement and maintain a policy such as increased PE than integrating physical activity into the existing curriculum. Testing these models would ideally require a well-implemented intervention with sufficient resources and duration to allow effects to accumulate over time. Unfortunately, many school-based physical activity interventions suffer from low implementation success [18], which challenges interpretation of the trial. Further, researcher-initiated school-based studies are rarely followed for more than 1 year, and typically dissipate once the study has ceased. Hence, the long-term benefits of the interventions are often unknown. Accordingly, natural experiments—a study design where a “naturally” occurring (i.e., not researcher-initiated) change in policy or behavior is utilized to study the impact of the change on relevant outcomes [19]—also offers benefits to understand the influence of physical activity in academic outcomes. Examples of natural experiments include changes in the build-environment to study the role of neighborhood safety and physical activity or the impact of targeted taxation on the consumption of sugar-sweetened beverages. In contrast to highly controlled interventions, natural experiments are independent of support from researchers, which facilitates both external validity and sustainability of the policy change. On the other hand, exposure in natural experiments is rarely randomly assigned, which calls for careful consideration of context-specific confounding and other biases. 

The aim of this study was to conduct a secondary analysis of an existing observational dataset that examined the effect of a trebling PE (i.e., 270 min/week) compared to the national standard of 90 min/week. The trebling condition consisted of curricular PE for six weekly lessons provided for 3–7 years during primary school. As such, this re-analysis examined the relationship of increasing curricular PE on academic performance upon the completion of compulsory school. Such an understanding of the relationship of greater amounts of PE on academic achievement are important for understanding the role of physical activity on children’s education and learning.

## 2. Materials and Methods

### 2.1. Setting and Study Design

This study is a secondary analysis from the Childhood Health, Activity, and Motor Performance School Study Denmark (CHAMPS-study DK, ClinicalTrials.gov Identifier: NCT03510494), a natural experiment implemented as a controlled intervention study including children from 10 public schools in the municipality of Svendborg, Denmark. At baseline in 2008, children in the first year of school (kindergarten) to the 4th grade were invited to participate in the study. The CHAMPS-study DK was set up to evaluate a range of outcomes stemming from a trebling of curricular PE (270 min per week distributed across at least three school days), initiated by the municipality of Svendborg (i.e., not researcher-initiated) [20]. Six of 19 schools in the municipality were willing and able to fund the additional PE classes and became “sports schools”. Four schools, matched on size, rural/urban and sociodemographic uptake area, agreed to serve as controls (“normal schools”). The additional PE was provided to all students from kindergarten to the 6th grade. Accordingly, from the 7th to 9th grade (final year of compulsory education in Denmark) the standard two PE lessons (90 min) per week were provided both at sports and normal schools. Hence, sports-school participants could receive additional PE for three (4th grade in 2008) to seven (Kindergarten in 2008) years. In 2014, a national school reform was implemented with one component of this reform mandating that schools provide students with 45 min of daily physical activity. All schools were free to decide how they implemented the 45 daily minutes, which should be in addition to existing recess and breaks. The CHAMPS-study DK was approved by the ethics committee of the region of Southern Denmark (S-20080047 and S-20140105) with written informed consent obtained from a parent or legal guardian and verbal consent from the participating children.

### 2.2. Participants

In 2008, all children from the 10 schools (N = 1507) were invited to participate in the study. Additional recruitment was pursued at subsequent follow-up waves each 6 months until 2010 [21,22]. We allowed ongoing recruitment of children moving into the school catchment area and of children who did not consent at study baseline. Of the 1305 children enrolled during 2008 to 2010, we did not retrieve exam data from 42 participants and had no information on parental education from 2 participants, leaving a total of 1261 students for analysis (see flowchart in Figure 1). As a secondary analysis, the analytical sample was expanded to include children enrolled at phase 2 of the CHAMPS-study DK conducted in 2012/2013 [23]. Phase 2 of the CHAMPS-study DK was an extension of the original study, recruiting both CHAMPS-I participants as well as children not previously enrolled in the study. This increased the eligible sample to 1940 children, with 1587 providing consent and all relevant data. We defined this as a secondary analysis because PE exposure during the initial phase of the study was unknown for a subset of participants enrolled at this stage. For these participants, exposure status was assigned based on current school attendance rather than school attendance at baseline. 

### 2.3. Content of the PE Program

In addition to increased PE, all PE teachers (in Denmark PE is mainly taught by PE specialists) attended a 40-lesson skill-developing course based on an Age-related Training Concept developed by the Danish organization for elite sports (Team Denmark) [24]. The purpose of this program is to augment development of body and motor skills in children and adolescents by considering their physical, physiological, mental and social development. In brief, the program was based on play, exercise and games with an increased focus on technical and coordinative skills in adolescence. Lessons were delivered 3 times per week. The additional PE was added to the existing curriculum and did not replace other academic subjects. Normal schools maintained national guidelines. The additional PE did not materialize in higher total activity levels at sports schools compared with normal schools, but school-time activity levels were higher at sports schools explained by more time spent in moderate and vigorous activity [25]. Further, the additional PE resulted in a more favorable cardiometabolic risk profile at sports schools 2 years after baseline, with this benefit being more pronounced among those with the least favorable profile at baseline [21,26]. This difference was not maintained at the long-term evaluation 6 years after baseline [27]. Additional details of the PE content and delivery at sports schools can be found elsewhere [20,28].

### 2.4. Outcomes 

Academic performance was obtained from the Academic Achievement registry [29], which contains grades from exams completed at the end of compulsory education (9th grade) in Denmark. Compulsory education was completed in 2014 by the oldest age-cohort and in 2018 by the youngest age-cohort (see Figure 2). Compulsory education in Denmark is completed by a series of 10 exams conducted by the Danish Ministry of Education, including fixed academic subjects: Danish language (4 exams), English oral language, written mathematics (2 exams) and science. In addition, one randomly selected subject from humanities (French language, German language, History, Christianity, English written language, social studies) and one randomly selected subject from science (Biology, physics/chemistry, geography, PE, mathematics oral presentation) are included in the exam battery. Oral exams (2 exams in Danish language, English oral language and science) are graded by a teacher and an external evaluator, while written exams (2 exams in Danish language, both exams in mathematics) are standardized achievement tests graded by an external evaluator. Randomly selected subjects can be oral or standardized tests. Our primary outcome was the combined academic performance (i.e., grade) of the 10 subjects calculated following the procedure of the Danish Ministry of Education [30]. Information on all 10 exams was required for inclusion in the primary analysis. As secondary outcomes, we included performance on subject-specific analyses if at least one fixed exam was completed and could be linked with our dataset. The Danish grading scale assigns one of the 7 possible numerical values, −03, 00, 02, 4, 7, 10, or 12. The grades 02, 4, 7, 10, and 12 represent passing grades, while failing grades are −03 and 00.

### 2.5. Additional Exposure Variables

Education attainment and household income for participant’s legal guardians were used as markers of socioeconomic status and extracted from Statistics Denmark [29,31] through linkage with the participants unique personal identification number (civil personal registration number). We used the highest completed education of legal guardians through 2020 as a marker of educational attainment. Education was categorized into 4 levels based on the International Standard Classification of Education (ISCED) as ISCED 0–2 (primary or lower secondary education), ISCED 3 (general upper secondary education and vocational upper secondary educations), ISCED 5–6 (short-cycle tertiary, medium-length tertiary and bachelor’s-level educations or equivalent), or ISCED 7–8 (second-cycle, master’s-level or equivalent educations, PhD-level educations). There is no ISCED level 4 in Denmark. Household income was expressed as quartiles of the mean yearly equivalized income from 2008 to the year the participant completed compulsory education. Household equivalized income was used because this metric accounts for differences in the number of family members of the household. If household income was discordant for the female and male legal guardian, they did not share households and we used the highest income reported. Height and weight were measured by trained research staff in 2008 according to a standardized protocol. We calculated body mass index as body weight (kilograms) divided by height squared (meters). We used aggregated exam results from the three years prior to initiation of the natural experiment to control for difference in education performance explained at the school level.

### 2.6. Statistics

Descriptive statistics are summarized using means with standard deviations for continuous variables and frequencies for categorical variables. Linear mixed regression models were used to contrast sports schools with normal schools for primary and secondary outcomes. Because the study is based on a natural experiment with schools selecting into the sports schools, group assignment was nonrandom. Further, we were also unable to collect academic performance data prior to group assignment because this data is not routinely collected in preschool children in Denmark. Given these concerns, we therefore controlled our models for age, gender, highest completed education of legal guardians and mean equivalized household income as fixed effects individual factors. To control for potential baseline imbalances in school-level academic environment, each participant was assigned the averaged 9th grade exams obtained at his/her school the 3 years prior to initiation of the sports schools (exams completed from 2006 to 2008). Finally, the models included cluster-indicators to account for nesting within schools. We used the school where the student attended the exam as a random effect (cluster-identifier) because this grouping explained more variance in our primary outcome than using baseline school as cluster-indicator. We repeated our analyses with stratification by gender and by exam year as a marker of different years of exposure to the additional PE. Finally, to provide sensitivity analyses, we repeated the analysis of the combined exam grade using two different assumptions for individuals with missing exams (and hence not included in the primary analysis of the combined grade). Sensitivity analysis 1 estimated the pooled grade as the mean exam score of available grades (e.g., if mathematics is missing, the pooled exam score is calculated as the mean of Danish, English language and Science). At least one exam grade was needed to be included in this analysis. Sensitivity analysis 2 included all participants by assigning missing exams a grade of −03 and recalculating the pooled exam grade. All statistical models satisfied distributional assumptions. Analysis was conducted using Stata version 16 (StataCorp, College Station, TX, USA) with a two-sided alpha of 0.05. 

## 3. Results

We included 1201 participants in the analysis of our primary outcome with 691 and 510 students from sports and normal schools, respectively. Baseline characteristics were similar at sports and normal schools (Table 1). The primary analysis revealed that children enrolled in sports schools performed a fifth of a grade higher on their combined exam score compared to children enrolled in normal schools (0.20 (95% confidence interval (CI): −0.12 to 0.52)). Results for exam-specific analyses were similar in magnitude with no results achieving statistical significance (Figure 3). This pattern was also consistent when stratified by gender (Table 2) and when we expanded the analytical sample to include participants enrolled during phase−2 of the study (ESM Table A1). Finally, results stratified by age-cohort did not indicate a dose-response based on years of exposure to additional PE (ESM Table A2). Our sensitivity analyses on the pooled exam performance, including children with missing exam grades, yielded similar results to the primary analysis based on complete cases, scenario 1: 0.19 (−0.14 to 0.52), *n* = 1261, scenario 2: 0.09 (−0.30, 0.49), *n* = 1267. 

The national school reform mandates 45 min of physical activity on each school day (not including recess and breaks). Implementation of the 45 min is managed at the school level. KG: Kindergarten, PE: Physical education, NSR: national school reform. Main outcome is 9th grade exams (end of compulsory education).

Adjusted for: individual-level factors: age, gender, parental education, household income, school where the exam was taken (random effect). School-level factors: aggregate exam grades 3 years prior to initiation of the natural experiment.

Pooled grade is average of Danish written language; Danish oral language; Mathematics, written; English, oral; Science + one random exam from humanities + one random exam from sciences.

## 4. Discussion

In a secondary analysis of the CHAMPS-study DK we found no evidence that the local initiative to treble curricular PE (i.e., 270 min a week) for 3–7 years, manifested in higher exam performance at sports schools upon completion of compulsory education (i.e., 9th grade) compared to schools that continued standard practice (i.e., 90 min a week). This observation was reiterated in gender-stratified analysis, when stratified by years of exposure to additional PE as a dose-response measure, in analyses using more lenient inclusion criteria, and under different sensitivity analysis assumptions.

The CHAMPS-study DK was not designed specifically to improve academic performance but had the overall aim of improving physical health and motor performance of the participants [20]. Further, the trebling of weekly PE did not result in greater overall physical activity levels of children attending sports schools [25]. In the short term, children at sports schools had higher in-school, but lower leisure-time, moderate-to-vigorous physical activity compared with children attending normal schools [25]. However, in line with the short-term benefits on metabolic risk markers and adiposity [21,26], we were unable to demonstrate that these differences in physical activity patterns were sustained over time [27]. Children and adolescents with higher levels of physical activity or cardiorespiratory fitness have higher academic performance and perform better on some cognitive tests, particularly those involving higher levels of executive functions (i.e., inhibition, working memory and cognitive flexibility) [5,32,33]. Maintaining physical activity of sufficient intensity and duration to increase cardiorespiratory fitness or prevent excess weight gain is a plausible factor linked with greater brain health [5,32]. Improvements in this factor may thus be needed to enhance academic performance.

An important strength of this study is the well-established implementation of the additional PE lessons. Previous research from our group demonstrated that all schools managed to schedule and complete the extra PE lessons all years [34]. However, because a trebling of PE did not manifest in higher total activity levels [25], these data cannot be used as evidence against the role of physical activity for improving academic performance. Rather, these results should be used to guide future interventions and, most importantly, because children were followed for the majority of their compulsory education, provide a unique opportunity to evaluate the potential benefits of a similarly framed policy on academic performance. This is in contrast to most researcher-initiated studies within the school system, which are rarely followed for more than 1 year. A second strength is the high level of participation at both sports and normal schools, which protects against selection bias. Because the additional PE was not randomized, but schools opted in as sports schools, we cannot refute residual biases. We used four matched schools for comparison and also controlled our models for factors possibly explaining individual and school-level academic performance such as markers of parental socioeconomic status and the general school-level academic environment before initiation of the sports schools. However, we were unable to control for baseline academic performance at an individual level because this data does not exist. There are numerous potential health benefits of increasing total physical activity levels through the school system [1], but the totality of the evidence from randomized controlled trials seeking to improve academic performance through physical activity is inconclusive [2]. There are strong examples from studies successfully integrating physical activity into the curriculum with subsequent effects on tests of mathematics skills [9,11], but other well-designed trials with large sample sizes and demonstration of implementation efficacy have failed to show between-group differences on academic outcomes [10,14]. Therefore, it is possible that in order to have an effect on academic performance, school-based physical activity interventions should either be intensive enough to increase overall physical activity levels or cardiorespiratory fitness of the participants, or integrate physical activity into the academic curriculum. Identifying and testing the “real-world” translation of putative mechanisms should remain an important research priority to advance understanding of the potential role of physical activity in promoting academic attainment. These questions cannot be reliably answered by observational research because of the strong socioeconomic gradients for physical activity, physical fitness and academic achievement.

We highlight the following limitations of this study. (1) The additional PE was discontinued after the 6th grade, meaning there were no differences in the amount of PE provided at sports and normal schools during the three years prior to assessment of academic performance (collected in the 9th grade). The impact of additional PE sustained or provided during secondary school on academic performance is therefore unknown. We note that an earlier evaluation of the sports schools on academic performance, assessed by standardized academic tests in Danish and Mathematics in the 3rd and 6th grade, showed no differences between sports and normal schools [13]. (2) Because a national school-reform was implemented in 2014 and one component of this reform was the obligation for schools to provide students with 45 min of daily physical activity [35], variations in physical activity between sports and normal schools may have been reduced. Because the additional PE was initiated by the local municipality and not controlled by researchers, we were also unable to control or monitor other “interventions” (e.g., diet campaigns) initiated by schools. (3) The content of the PE lessons at sports and normal schools was, besides the guidelines of the Age-related Training Concept, planned and executed by each school/teacher. The content was therefore heterogeneous, making it difficult to pinpoint potentially salient features of academic performance enhancing PE. (4) The CHAMPS-study DK included only a small number of schools and the PE assignment was not randomized. We therefore cannot eliminate confounding from factors predicting selection into sports schools. We controlled for markers of socioeconomic factors, and baseline characteristics were similar across study arms. (5) Due to the lack of baseline measure for the primary outcome, we were unable to examine potential disproportionally greater benefits of the additional PE among those with the lowest academic performance at baseline. Earlier studies have suggested more pronounced benefits in this group [14].

In conclusion, a well-implemented trebling of curricular PE until the 6th grade in a natural experiment conducted within a single municipality did not manifest in superior exam performance of sports-school children, compared to normal schools, upon completion of compulsory education in the 9th grade.

## Figures and Tables

**Figure 1 children-08-00057-f001:**
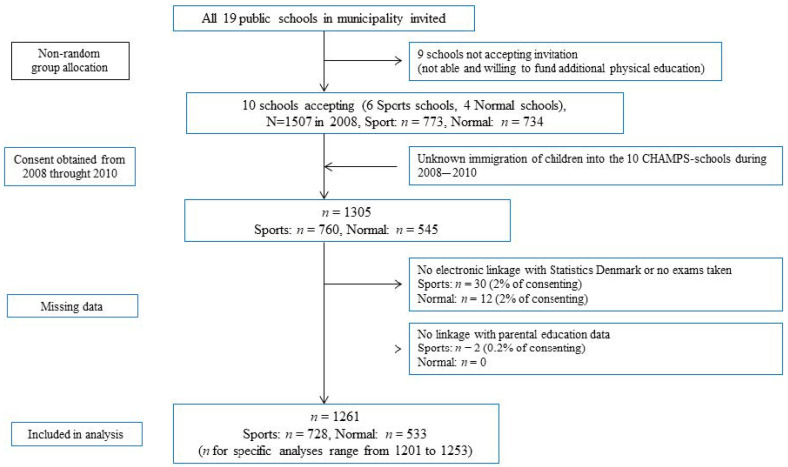
Participant flowchart.

**Figure 2 children-08-00057-f002:**
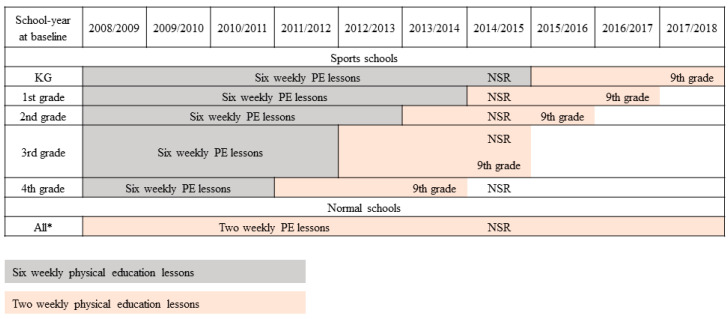
Years of physical education exposure and final year of compulsory education in the Childhood Health, Activity, and Motor Performance School Study Denmark (CHAMPS-study DK) cohort by school-year at baseline and PE exposure status (2008–2018). * Age-cohorts following same pattern as sports schools. KG: kindergarten, PE: physical education, NSR: national school reform.

**Figure 3 children-08-00057-f003:**
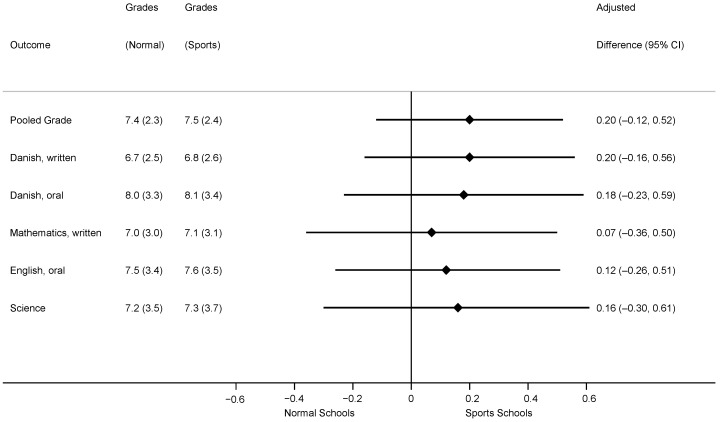
Difference in exam performance at end of compulsory education.

**Table 1 children-08-00057-t001:** Baseline characteristics.

	Normal Schools (*n* = 533)	Sports Schools (*n* = 728)
Age at exam (years)	16.0 (0.3)	16.0 (0.4)
Girls (*n*, %)	261 (49)	401 (55)
Body mass index in 2008 *	16.5 (2.0)	16.4 (2.1)
**Highest Parental Education (*n*, %)**		
ISCED 0–2	5 (1)	21 (3)
ISCED 3	186 (35)	288 (40)
ISCED 5–6	286 (54)	324 (45)
ISCED 7–8	56 (11)	95 (13)
**Household Income (*n*, %)**		
Lowest	94 (18)	153 (21)
Second Lowest	134 (25)	164 (23)
Second Highest	142 (27)	191 (26)
Highest	163 (31)	220 (30)
**School-level factors**	Normal Schools (*n* = 4)	Sports Schools (*n* = 6)
Average 9th grade exams 2005–2008	6.3 (0.5)	6.4 (0.6)

Age defined as difference between birthdate and May 30th the year of graduation. International Standard Classification of Education (ISCED) 0–2: Primary or lower secondary, ISCED 3: General upper secondary + vocational upper secondary, ISCED 5–6: Short cycle tertiary, medium length tertiary and bachelor’s level, ISCED 7–8: second cycle master’s level, and PhD, ISCED 4 is not used in Denmark (Andersen et al., 2017-PMID: 28893221). Household Income: average of equivalized household income from 2008 through 9th grade graduation year. * *n* sports schools = 634, normal schools = 483.

**Table 2 children-08-00057-t002:** Difference in exam scores, stratified by gender.

	Normal Schools	Sports Schools	Adjusted Mean Difference Using Normal Schools as Reference	*p*-Value
**Girls**				
Pooled Grade	7.6 (2.3), *n* = 253	7.7 (2.3), *n* = 377	0.24 (−0.21, 0.69)	0.29
Danish written language	7.3 (2.5), *n* = 260	7.4 (2.5), *n* = 398	0.26 (−0.22, 0.74)	0.30
Danish oral language	8.8 (3.1), *n* = 261	8.9 (3.1), *n* = 399	0.32 (−0.16, 0.79)	0.19
Mathematics, written	6.8 (2.9), *n* = 259	6.6 (3.2), *n* = 398	−0.28 (−0.85, 0.30)	0.35
English, oral	7.5 (3.4), *n* = 258	7.7 (3.5), *n* = 389	0.26 (−0.41, 0.93)	0.45
Science	7.8 (3.4), *n* = 256	7.5 (3.7), *n* = 389	−0.22 (−0.87, 0.43)	0.51
**Boys**				
Pooled Grade	7.1 (2.3), *n* = 257	7.2 (2.5), *n* = 314	0.12 (−0.30, 0.55)	0.57
Danish written language	6.1 (2.5), *n* = 268	6.1 (2.5), *n* = 326	0.05 (−0.45, 0.54)	0.85
Danish oral language	7.2 (3.3), *n* = 267	7.1 (3.5), *n* = 326	−0.17 (−0.70, 0.37)	0.54
Mathematics, written	7.3 (3.0), *n* = 269	7.5 (3.0), *n* = 324	0.39 (−0.18, 0.96)	0.18
English, oral	7.5 (3.5), *n* = 268	7.5 (3.6), *n* = 319	0.00 (−0.65, 0.66)	0.99
Science	6.6 (3.5), *n* = 265	7.1 (3.7), *n* = 321	0.54 (−0.10, 1.18)	0.10

Adjusted for: individual-level factors: age, gender, parental education, household income, school where the exam was taken (random-effect). School-level factors: aggregate exam grades 3 years prior to initiation of the natural experiment. Pooled grade is an average of Danish written language; Danish oral language; Mathematics, written; English, oral; Science + one random exam from humanities + one random exam from sciences.

## Data Availability

Restrictions apply to the availability of these data. Data was obtained from the Danish National Archives and Statistics Denmark and are available at https://www.sa.dk/en/ and https://www.dst.dk/en.

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
