# Peer review of "Does Additional Physical Education Improve Exam Performance at the End of Compulsory Education? A Secondary Analysis from a Natural Experiment: The CHAMPS-Study DK"

_children, 2021, doi:10.3390/children8010057_

Round 1

Reviewer 1 Report

Some items , such as grades of the school attendants, exercises protocol, could be explained clearly, for better understanding of an international reviewer.

Author Response

Some items , such as grades of the school attendants, exercises protocol, could be explained clearly, for better understanding of an international reviewer.

Thank you for this comment. The sports schools received additional physical education delivered by physical education (PE) teachers. PE teacher also delivered PE at normal schools as this is standard practice in Denmark. In addition, PE teachers at sports schools were given a 40-lessons skill course based on an age-related training concept. This concept is based on the provision of bodily and motor developing PE, which is aligned with the abilities of participating children. We detail this in lines 159-178. As such, the program is generic and there is no record of what activities the children may have actually performed, as this was planned and executed locally.

We have modified the description of the grading system in line 203:

‘The Danish grading scale assigns one of the 7 possible numerical values, -03, 00, 02, 4, 7, 10, or 12. The grades 02, 4, 7, 10, and 12 represent passing grades while failing grades are -03 and 00.’

Reviewer 2 Report

The document presents an interesting topic and the authors did a good work. Generally, it is a good paper but to improve some sections, I leave some comments below:

General comments:

Align the whole article on the left.

In general, throughout the article, place all tables and figures in the order in which they appear in the article.

Please check the tables and figures you refer to in the text, it seems you name some but you pretend to refer to others.

Introduction

Line 51. Do you would to say “caries” or “ carries”?

Materials and methods

Line 143. I think it is Figure 1, instead of Figure 2. Change it please.

If in figure 1, it is indicated that the participants included in the final sample are 1261 participants, comment on the participants' paragraphs this as well. Perhaps the paragraph in lines 256-259 could be passed: "Of the 1305 children enrolled during 2008 to 2010, we did not retrieve exam data from 42 participants and had no information on parental education from 2 participants, leaving a total of 1261 students for analysis (Figure 1).

Lines 143-149. It is not entirely clear the sample of phase 2 conducted in 2012/2013. How much would the total sample add up to in this second phase? Please, explain better in this section and in Statistics the phase 2 of the study.

Results

Explain the results in the order the tables appear. For example, lines 262-265 ("The primary analysis revealed that children enrolled in sports schools performed a fifth of a grade higher on their combined exam score compared to children enrolled in normal schools (0.20 264 [95% confidence interval (CI): -0.12 to 0.52]). ") refer to Table 3. Place them after the rest, or change the numbering table 3.

Line 267. Do you refer to figure 2 or figure 3?

If you mention figure 3 before table 2, place it in the text in this order.

Conclusion

Indicate that this conclusion it is in your study, referring to city/country, not in general.

Author Response

General comments:

Align the whole article on the left.

Thank you for this comment. The manuscript has been formatted by the editorial office.

In general, throughout the article, place all tables and figures in the order in which they appear in the article.

Thank you for this comment. The manuscript has been formatted by the editorial office.

Please check the tables and figures you refer to in the text, it seems you name some but you pretend to refer to others.

Thank you for alerting us to these errors. We have changed the manuscript to refer to the correct figures. 

Line 145: reference to Figure 2 changed to Figure 1.

Line 276: reference to Figure 2 changed to Figure 3.

Introduction

Line 51. Do you would to say “caries” or “ carries”?

Thank you for this correction. We meant to say ‘carries’.

Materials and methods

Line 143. I think it is Figure 1, instead of Figure 2. Change it please.

We have changed accordingly.

If in figure 1, it is indicated that the participants included in the final sample are 1261 participants, comment on the participants' paragraphs this as well. Perhaps the paragraph in lines 256-259 could be passed: "Of the 1305 children enrolled during 2008 to 2010, we did not retrieve exam data from 42 participants and had no information on parental education from 2 participants, leaving a total of 1261 students for analysis (Figure 1).

Thank you for this suggestion. We have modified as suggested in line 141 and, as a result, also made changes in line 265.

Lines 143-149. It is not entirely clear the sample of phase 2 conducted in 2012/2013. How much would the total sample add up to in this second phase? Please, explain better in this section and in Statistics the phase 2 of the study.

We have added additional information in line 149:

‘Phase 2 of the CHAMPS-Study DK was an extension of the original study, recruiting both CHAMPS-I participants as well as children not previously enrolled in the study. This increased the eligible sample to 1940 children with 1587 providing consent and all relevant data.’

Results

Explain the results in the order the tables appear. For example, lines 262-265 ("The primary analysis revealed that children enrolled in sports schools performed a fifth of a grade higher on their combined exam score compared to children enrolled in normal schools (0.20 264 [95% confidence interval (CI): -0.12 to 0.52]). ") refer to Table 3. Place them after the rest, or change the numbering table 3.

The first results presented in line 271 (‘The primary analysis revealed that children enrolled in sports schools performed a fifth of a grade higher on their combined exam score compared to children enrolled in normal schools (0.20 264 [95% confidence interval (CI): -0.12 to 0.52])’ ) refers to Figure 3 and we mention this information in the ensuing sentence which has now been corrected to refer to the correct figure ‘Results for exam-specific analyses were similar in magnitude with no results achieving statistical significance (Figure 3)’.

Next, we reference Table 2 in line 276 (‘This pattern was also consistent when stratified by gender (Table 2)’). We agree the published version should show Figure 3 before Table 2 and will arrange for this in agreement with the editorial office.

Line 267. Do you refer to figure 2 or figure 3?

We have corrected the text to refer to Figure 3.

If you mention figure 3 before table 2, place it in the text in this order.

Thank you for this comment. The manuscript has been formatted by the editorial office. We fully agree that Figure 3 should be placed before Table 2.

Conclusion

Indicate that this conclusion it is in your study, referring to city/country, not in general.

We have modified line 441 as:

‘In conclusion, a well-implemented trebling of curricular PE until the 6th grade in a natural experiment conducted within a single municipality did not manifest in superior exam performance of sports school children, compared to normal schools, upon completion of compulsory education in the 9th grade.’

Round 2

Reviewer 2 Report

Thank you for taking my comments on board, and responding thoroughly, the manuscript has evolved and improved as a result. 

Best wishes with your research.